# Trends and Geographical Distribution of Family Health Strategy in Brazil from 2009–2023

**DOI:** 10.3390/healthcare13111246

**Published:** 2025-05-26

**Authors:** Pedro Henrique Sales Barbosa, Bárbara Sarni Sanches, Maria Luisa dos Anjos Correa do Espírito Santo, Hudson Pabst, Marcelo Gerardin Poirot Land, Heitor Siffert Pereira de Souza

**Affiliations:** 1Departamento de Clínica Médica, Faculdade de Medicina, Universidade Federal do Rio de Janeiro, Rio de Janeiro 21941-913, Brazil; phsalesbarbosa@gmail.com (P.H.S.B.); barbarasarni@hotmail.com (B.S.S.); mluisaanjosesanto@gmail.com (M.L.d.A.C.d.E.S.); land.marcelo@gmail.com (M.G.P.L.); 2Secretaria Municipal de Saúde do Rio de Janeiro, Rio de Janeiro 20211-110, Brazil; 3Instituto de Ciência e Tecnologia em Biologia do Câncer e Oncologia Pediátrica (INCT BioOncoPed), Porto Alegre 90035-903, Brazil; 4Departamento de Hematologia Pediátrica, Instituto de Puericultura e Pediatria Martagão Gesteira, Rio de Janeiro 21941-912, Brazil

**Keywords:** public health, primary health, family health strategy, Brazilian public health system, Sistema Único de Saúde, human development index

## Abstract

**Background/Objectives**: The Brazilian Unified Health System was established in 1988 as a public health system with principles of universality, equity, and integrality. One of Brazil’s main strategies to strengthen universal healthcare is the Family Health Strategy (FHS), a primary health care policy established in 1994 and fully incorporated in the country in 2006. This study aims to describe the time trends of FHS coverage in Brazil and its states from 2008 to 2023 and to correlate this coverage with the states’ Human Development Index (HDI). **Methods**: Data on the number of FHS teams, population, and HDI during the period were collected for each Brazilian state in the Brazilian Ministry of Health’s public-access databases. Prais–Winsten regression was used to conduct a time series analysis for each state and country. The annual percentage change (APC) was used to describe time trends in time series. Linear regression was used to identify a correlation between HDI and FHS coverage across states. **Results**: The FHS coverage in the country increased from 66.81% to 84.66% from 2009 to 2023. Disparities in coverage between regions are evident throughout the entire study period. The Northeast region (NE) exhibited higher FHS coverage but lower APC rates compared to other regions. Results suggest a negative correlation between HDI and FHS coverage in all Brazilian states for 2009, 2012, 2015, 2018, 2021, and 2023. **Conclusions**: The FHS coverage increased in Brazil and its states during the period. The highest coverages were found in states from the North (N) and NE regions, and the lowest were in the Southeast region. Nine federative units achieved full coverage (100%) and maintained it afterward, with seven from the N and NE. A negative correlation was found between FHS and HDI, suggesting that the expansion of FHS effectively targets vulnerable populations.

## 1. Introduction

The Brazilian Unified Health System, also known as Sistema Único de Saúde (SUS), is a universal health system funded by the Brazilian government and established in 1988 with the advent of the new Brazilian Constitution [1]. The fundamental principles on which SUS is based include universality, equity, and integrality of care. SUS’s conception, creation, and development were propelled by intense popular requirements to guarantee citizen’s right to health. The Brazilian government is responsible for implementing measures to improve the population’s health status and combating social inequities. Despite the constitutional definition of a universal and cost-free public system, the implementation of SUS also involved integrating public and private institutions. In this sense, the universality principle guarantees access to healthcare services without discrimination. The equity principle aims to reduce established inequalities, treating every individual in a particular and equitable manner. Moreover, the integrality principle seeks to comprehend the individual and their biological, psychic, and social reality [2].

Over nearly 35 years, the expansion and development of SUS have resulted in a significant increase in health service coverage, addressing the nation’s healthcare needs [3]. One of the reasons for the widespread coverage of health services is primary health care (PHC), which represents the “entrance” to the health system and is conducted by the National Politics of Primary Health [4]. The PHC emerged in Brazil in accordance with the values established in the Declaration of Alma-Ata, which reinforces the importance of efforts from diverse societal sectors to achieve integral community health and combat social inequities. [5]. Additionally, PHC is guided by essential attributes, including first-visit care, integrality of care, longitudinal care, and coordination, as well as derived attributes such as family orientation, community orientation, and cultural competence [4,6].

The PHS in Brazil is mainly organized by the Family Health Strategy (FHS). FHS was implemented in 1994 and became part of the National Politics of Primary Health in 2006 [4,7]. FHS is a recognized model for PHC organizations, having expanded, qualified, and consolidated its attributes [8].

In Brazil, each family health team (FHT) is responsible for all individuals within a specific geographic area, known as a territory. The FHT organization differs from the traditional healthcare model, which is focused on the physician’s protagonism. In turn, the FHT consists of a generalist physician or family physician, a nurse, a nursing assistant, and community health agents; it may also include a dentist and dental assistant [7]. The community health agent plays an essential role in FHS among FHT members. The community health agent is a professional who resides in the community where the FHT operates and is responsible for connecting the community with health services, facilitating healthcare expansion, and ensuring continuity [9,10]. Therefore, this multifaceted organization of the FHT is a powerful tool for improving healthcare and providing access to a community territory, including its particular characteristics, dynamics, and vulnerabilities [8,11].

According to the Pan American Health Organization (PAHO), investment in PHC must be prioritized to achieve universal health in the Americas [12]. PAHO supports that the FHS has been the most effective organization form of PHC in Brazil, providing the best results in public health compared with other organization models, such as raising access to healthcare and increasing health indicators (reduction in hospitalization by PHC-sensitive conditions and reduction in infant, maternal, and preventable causes of mortality) [3,13].

Therefore, this study aims to update the analysis of FHS potential coverage in Brazil, encompassing its geographical particularities from 2009 to 2023, and to investigate whether the distribution of resources addresses inequity.

## 2. Materials and Methods

### 2.1. Study Design

We performed an analytical ecological study using data from public access data banks. Since no individual patient information was collected, the study was not submitted to an ethics committee.

### 2.2. Data Source

Data were retrospectively obtained from public access data banks, namely the Primary Health Information System (E-Gestor) (https://egestoraps.saude.gov.br/, accessed on 15 January 2025), the Health Informatics Department of the Brazilian Ministry of Health (DATASUS) (https://datasus.saude.gov.br/, accessed on 15 January 2025), the Brazilian Institute of Geography and Statistics (IBGE) (https://www.ibge.gov.br/, accessed on 15 January 2025), and the United Nations Development Programme (PNUD) (https://www.undp.org/pt/brazil, accessed on 15 January 2025). The total area population and the number of family health teams from the Brazilian public health system were obtained from each Brazilian region and state from 2009 to 2023. The Human Development Index (HDI) was obtained for each Brazilian federative unit in 2010 and 2021. The HDI measures a region’s overall development based on health, education, and income. It combines life expectancy, years of schooling, and gross national income per capita to reflect progress in these areas. HDI values range from 0 to 1, with higher values indicating greater human development and well-being.

### 2.3. Macro-Regions and Federative Units Division

Brazil is divided into 27 federative units (states), which are grouped into five different macro-regions, hereinafter referred to as regions. The North region (N) comprises the states of Acre, Amapá, Amazonas, Pará, Rondônia, Roraima, and Tocantins. The Northeast (NE): Alagoas, Bahia, Ceará, Maranhão, Paraíba, Pernambuco, Piauí, Rio Grande do Norte, and Sergipe. Midwest (MW): Goiás, Mato Grosso, Mato Grosso do Sul, and Distrito Federal. Southeast (SE): Espírito Santo, Minas Gerais, Rio de Janeiro, and São Paulo. South (S): Paraná, Rio Grande do Sul, and Santa Catarina.

### 2.4. Missing Values Imputation

Data on each states’ HDI were missing for 2009, 2011, 2022, and 2023 (26.6% missingness). To address this, we used the stochastic regression imputation method [14]. We compared different regression models (linear, logarithmic, inverse, quadratic, cubic, exponential, and logistic) and model complexity and R^2^ values as criteria to select the best-fitting model. Linear regression showed the best performance, with statistically significant imputations in most states (*p* < 0.05; R^2^ > 0.60). Therefore, linear regression was chosen as the preferred curve estimation model. This analysis was conducted in IBM SPSS Statistics Software, version 20.

### 2.5. Coverage Extension of Family Health Strategy Calculation

The coverage extensions of family health strategy calculation in a specific area were performed using the following formula [15]:Coverage extension (%)=(Number of FHS teams ∗3450 / Total Area Population) ∗100
where the number of FHS teams and the total area population are represented in absolute numbers and the coverage in percentage.

We used the simple average of the monthly coverage values to estimate coverage in a specific area over the course of one year.

### 2.6. Graphics, Images, and Tables Design

The graphics and images were designed using RStudio 2024.09.0+375 “Cranberry Hibiscus” Release for Windows.

### 2.7. Statistical Analysis

Time series analysis was conducted for each of the 27 Brazilian states using the Prais–Winsten estimation method, which is defined as follows:Yt=α+β∗Xt+ϵt
where Yt is the dependent variable, α is the intercept, Xt is the independent variable, and ϵt is the error term. This approach is used to estimate the correlation between errors at time t and t − 1, and these errors should follow a first-order autoregressive (AR (1)) process. So, the error term is defined as follows:ϵt=ρϵt−1+ut
where ρ is the autocorrelation parameter, and ut is white noise.

So, the Prais–Winsten formula for this study was as follows:FHS coverage=α+β ∗ year+ϵt

Based on this estimate, the outcome and predictor variables are transformed to eliminate the correlation from the error term when a linear regression model is applied to the modified data. This is made through the Cochrane–Orcutt transformation. After the data are corrected by autocorrelation, Prais–Winsten estimates the model parameters. The fitted values obtained for the FHS coverage were extracted from the model. This analysis used the “prais” package in RStudio version 2024.09.0+375.

For time trend analysis, the annual percentage change (APC) of the predicted FHS coverage was calculated using the approach proposed by Bottomley et al. [16], which is defined as follows:APC=[−1+10b1]∗100%

The value of b1 is the slope of the independent variable (year) when considering the outcome of the Prais–Winsten regression as a natural base logarithm. This analysis was also conducted in RStudio 2024.09.0+375 “Cranberry Hibiscus” Release for Windows.

The association between HDI and FHS coverage was evaluated through linear regression curve estimation for 2009, 2012, 2015, 2019, and 2023, considering the HDI as the independent variable and FHS coverage as the dependent one. This analysis was conducted in RStudio version 2024.09.0+375.

## 3. Results

### 3.1. Family Health Strategy Coverage in Brazil and Its Macro-Regions

We first conducted a time trend analysis of the FHS coverage in the Brazilian territory and its macro-regions from 2009 to 2023 (Figure 1). In 2009, the fitted FHS coverage in Brazil was 66.81% (CI = 95%; 65.95–67.67%), with nearly 30,415 FHT for 189,612,814 residents. Then, in 2023, this coverage rose to 84.66% (CI = 95%; 83.80–85.52%), with 50,005 FHT for 213,317,639 residents, showing an APC of 3.97% (CI = 95%; 3.35–4.58%) within the period.

Concerning the macro-regions, in 2009, the NE showed the highest coverage (86.20%; 95%CI: 85.69–86.71%), followed by the N (68.19%; 95%CI: 68.03–70.34%), the S (56.44%; 95%CI: 54.74–60.58%), the MW (48.46%; 95%CI: 46.30–50.63%), and the SE (46.72%; 95%CI: 45.50–47.93%). Data from 2023 showed that the NE remained with the highest coverage (98.21%; 95%CI: 97.70–98.72%), succeeded by the S (83.11%; 95%CI: 81.42–84.81%), the N (79.81%; 95%CI: 77.66–82.00%), the MW (78.21%; 95%CI: 76.04–80.38%), and the SE (71.19%; 95%CI: 70.00–72.40%). The FHS coverage through Brazilian states is represented in Figure 2. The APC through regions from 2009 to 2023 was 7.81% in MW (95%CI: 6.88–8.75%), 7.22% in SE (95%CI: 6.03–8.43%), 6.61% in S (95%CI: 5.26–7.97%), 2.66% in N (95%CI: 1.21–4.07%), and 2.17% in NE (95%CI: 1.88–2.46%).

### 3.2. Family Health Strategy Coverage in Brazilian Federative Unities

Analysis of FHS coverage in each federative unit also revealed that 18 federative units experienced a highly significant increase in FHS coverage from 2009 to 2023, with a notable emphasis on the Distrito Federal, which demonstrated the highest growth in the country (APC = 35.11, 95% CI: 30.41–39.98, *p* < 0.05). Amapá was the only federative unit that experienced a decrease in FHS coverage during the period (APC = −2.56, 95% CI: −4.63 to −0.44, *p* < 0.05). Moreover, Tocantins, Piauí, and Paraíba did not increase their total coverage after already achieving 100% FHS coverage in 2009. Although the coverage was not considered to be 100%, these federative units increased the number of FHTs and decreased the number of residents per FHT from 2009 to 2023. Finally, five federative units did not show a statistically significant increase in FHS coverage (Table 1).

### 3.3. Human Development Index Shows a Negative Correlation with Family Health Strategy Coverage

Linear regression analysis revealed a strong negative correlation between HDI values and FHS coverage across Brazilian states during the study period (Figure 3). In 2009, this correlation was stronger (β1 = −2.87, 95%CI: −4.25–−1.49, *p* < 0.05), gradually weakening over the years and becoming less pronounced in 2023, with the weakest association (β1 = −1.58, 95%CI: −2.94–−0.22, *p* < 0.05).

## 4. Discussion

In this study, we investigated the time trends and spatial distribution of FHS coverage in the Brazilian territory, its macro-regions, and states from 2009 to 2023. The results showed that the FHS coverage across the country increased over time from 2009 to 2023. Considering the country’s macro-regions, although all showed an increase in FHS coverage throughout the study period, coverage in the NE continued to be the highest, while in the SE, it was the lowest in the country. Regarding the federative units, more than 60% of them showed a significant increase in FHS coverage in the period.

Data showed that Brazil and all its macro-regions presented an enhancement in FHS coverage between 2009 and 2023. Nevertheless, such improvement substantially differs in each macro-region. The NE and N regions presented the smallest progression in FHS coverage according to the annual percentage change. This phenomenon needs to be carefully interpreted, as it can reveal some of the barriers these macro-regions had to overcome to achieve 100% coverage; however, it also represents the earlier expansion of FHS coverage in these areas. Seven of the eighteen states (38.8%) in these regions achieved 100% FHS coverage during the study period, with three of them (Tocantins, Piauí, and Paraíba) reaching this milestone since 2009. A coverage of 100% in the other nine states from different regions was observed only in Santa Catarina in 2023 (11.1%). Since we did not consider coverage numbers above 100%, N and NE might appear to be experiencing a lower improvement in FHS coverage, but the reason is that they have reached the maximum coverage earlier than other regions. To investigate quality improvement in different regions, it will be critically important to consider various parameters in future research, such as hospitalizations for Ambulatory Care Sensitive Conditions, home visits, and infant mortality [17,18]. In addition to the chronic underfunding of the health sector in Brazil, one of the main obstacles of the N and NE regions is the enormous area and low population density, which may render regional integration challenging to implement. Moreover, the heterogeneity in FHS coverage across states within a macro-region may contribute to the comparatively low regional growth rates. Compounding these issues is the scarcity of research and the poor quality of data available for these areas, which might hinder the ability to guide effective health policies [17,19]. Therefore, achieving a substantial increase in FHS coverage across a macro-region requires a concerted and combined effort from all subunits within that region.

Looking at the federative units, five showed no significant increase in FHS during this period. In fact, high FHS coverage was already detected in three federative units: Maranhão (100%), Rio Grande do Norte (100%), and Sergipe (98.35%). Otherwise, Amapá exhibited a negative tendency for FHS coverage. Further studies are needed to understand the specific characteristics of these regions, particularly those with decreased or non-significant increases in FHS coverage. Nevertheless, the expansion of FHS coverage and the broader health market in previously underserved areas has led to a relative shortage of physicians in Brazil over the past two decades. As the number of health facilities increased, the demand for medical professionals grew accordingly. However, physicians were often drawn to larger, more developed cities. To increase the number of doctors in underserved areas, particularly in rural and remote regions, the Brazilian government launched the “Mais Médicos” (More Doctors) program in 2013. Although the analysis of the effect of the program indicated progress toward a more equal distribution of physicians, a considerable difference in the level of equality in PCP distribution remains, especially across states [20]. Despite the controversy surrounding the launch of the More Doctors Program, improvements in FHS coverage may likely arise from regional incentives, such as the expansion and refinement of the program, for example [21,22]. However, a misdistribution of public investment may explain the differences in the enhancement of FHS through Brazilian federative units during previous years. Despite the efforts from the federal government and the National Politics of Primary Health Care, investment in FHS still depends on the administration of individual federative unit governors and city mayors.

The negative correlation between HDI and FHS corroborates the equity principle from SUS as it highlights a higher coverage in more vulnerable areas. According to the constitution, the Brazilian government is obligated to develop a health policy aimed at reducing health inequities among the population [1]. These interventions aim to prevent the consequences of the inverse care law and the inverse equity hypothesis. These theories suggest that the availability of medical care and health technologies tend to vary inversely with the population’s need for it [23,24]. Additionally, healthcare inequities often occur due to economic factors. Therefore, less FHS coverage may mean not only fewer physicians but also less access to follow-up exams, disease prevention, and longitudinal healthcare. As shown by data from the National Health Research, high-HDI areas tend to present a higher prevalence of private health services [25]. In this sense, the Brazilian government is making efforts to reduce health inequity phenomena while investing in PHC, especially in FHS, in low-HDI areas [11,26].

Previous studies have analyzed FHS coverage in Brazil before the COVID-19 pandemic [15,18] using various methodologies. In this sense, the COVID-19 pandemic highlighted the importance of PHC in combating pandemics and public health emergencies, as many patients received their first medical care in PHCs [27,28]. Analyzing data from 2006 to 2016, Neves et al. utilized databanks from e-SUS Atenção Básica (e-SUS AB) and reported that 13 federative units demonstrated FHS coverage below 75% in 2016 [15]. Conversely, the current study found FHS coverage below 75% in only six federative units, suggesting a potential upward trend in FHS coverage nationally. Giovanella et al. used data collected individually from National Health Research (PNS) participants in 2013 and 2019, and then compared these data across the two years [18]. Some vulnerability-related variables, such as educational level and family income, showed a negative correlation with personal enrollment in family health teams. None of these studies correlated the FHS coverage with HDI to compare social inequities and populational access to PHC. Also, no negative growth of FHS was detected in Amapá during the studied period. Additionally, Andrade et al. presented a quali-quanti analysis of the patterns of FHS coverage expansion in Brazilian municipalities between 1998 and 2012 pointing to a heterogeneous expansion mainly influenced by funding mechanisms, availability of private insurance, and population size [29].

The results of this study show the geographic distribution and temporal trends of FHS coverage in one of the largest populations of a developing country based on data continuously updated in a single database (DATASUS). Despite the consistency and quality of data collection, some limitations must be acknowledged. First, this is a retrospective study, which limits the investigation to the information registered in the database, including all the variables investigated. Therefore, we cannot rule out the likelihood that essential variables may not have been included in this study. Moreover, data collected from DATASUS do not account for possible disparities in access to and the quality of healthcare in less developed regions, which may result in potential discrepancies. Furthermore, this study cannot accurately determine the real percentage of the population assisted by the FHS when the number of people assisted by an FHT is lower than 3450. Finally, it is well known that the traditional model of FHT cannot provide high-quality health assistance to some minority groups, such as homeless people, persons deprived of liberty, and riverine communities. These minorities demand alternative types of healthcare teams for comprehensive health assistance [30,31,32]. Due to these limitations, the results of this study may not accurately reflect the quality of healthcare provided by PHC to the entire population. Disparities in the number of patients per FHT can mean that the FHS coverage reported for some areas may not accurately reflect the percentage of people receiving health assistance. Instead, these figures could reflect the potential capacity of the FHS within those areas.

## 5. Conclusions

This study highlights the expansion and distribution of FHS coverage in Brazil between 2009 and 2023, underscoring a nationwide increase alongside significant regional variations. While the Northern and Northeastern regions achieved the highest coverage, reaching 100% in several states, they exhibited slower growth rates because they had already attained full coverage. A negative correlation between FHS coverage and the HDI suggests that the program expanded most significantly in lower HDI (and, therefore, underserved) areas, improving access to healthcare in those regions. This aspect demonstrates that the policy effectively targeted vulnerable populations, aligning with Brazil’s public health equity goals. However, despite overall progress, critical issues such as physician distribution, funding inconsistencies, and data limitations suggest that reported coverage may not always reflect actual healthcare access and quality for all populations, including minority groups.

## Figures and Tables

**Figure 1 healthcare-13-01246-f001:**
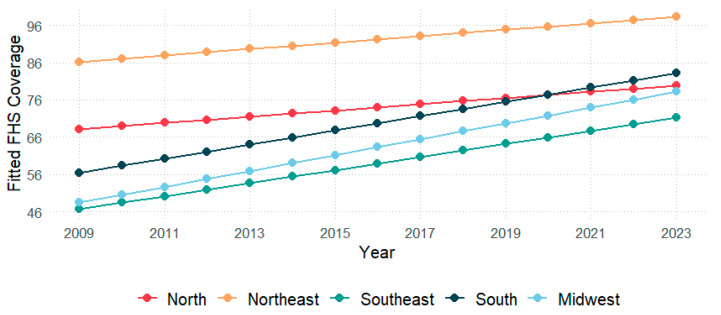
Fitted values of Family Health Strategy coverage by macro-region in Brazil from 2009 to 2023.

**Figure 2 healthcare-13-01246-f002:**
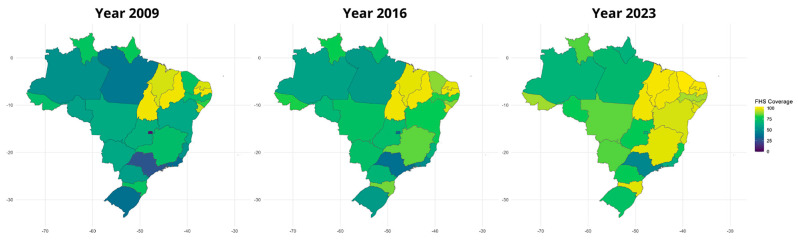
Spatial distribution of Family Health Strategy coverage in Brazil between 2009–2023.

**Figure 3 healthcare-13-01246-f003:**
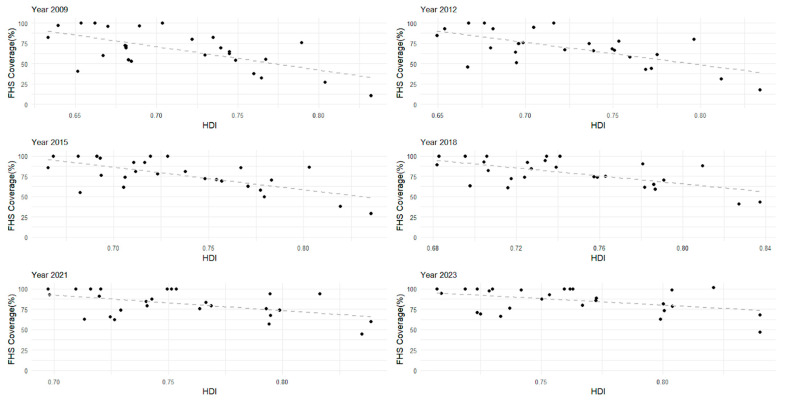
The linear trend for estimating the correlation between the Human Development Index (HDI) values of federative units (x) and their Family Health Strategy (FHS) coverage (y) for the years 2009, 2012, 2015, 2018, 2021, and 2023.

**Table 1 healthcare-13-01246-t001:** Estimated Family Health Strategy coverage (%) in federative units in 2009, 2016, and 2023, their annual percentage change (APC), and their *p*-value considering a confidence interval of 95%.

Federative Unit	2009 (%)	2016 (%)	2023 (%)	APC (%)	*p*-Value
Rondônia	58.12	68.26	78.40	5.27	<0.05
Acre	71.82	81.00	90.18	3.97	>0.05
Amazonas	51.51	58.97	66.43	4.25	<0.05
Roraima	75.32	79.67	84.01	1.72	>0.05
Pará	40.87	55.51	70.15	9.32	<0.05
Amapá	77.13	71.37	65.60	−2.56	<0.05
Tocantins	100.00	100.00	100.00	0	<0.05
Maranhão	96.78	98.56	100.00	0.70	>0.05
Piauí	100.00	100.00	100.00	0	<0.05
Ceará	73.37	88.13	100.00	5.8	<0.05
Rio Grande do Norte	96.70	98.79	100.00	0.70	>0.05
Paraíba	100.00	100.00	100.00	0	<0.05
Pernambuco	72.13	81.60	91.04	3.87	<0.05
Alagoas	82.01	88.00	94.00	2.25	<0.05
Sergipe	94.39	96.37	98.35	0.67	>0.05
Bahia	60,78	78.54	96.30	7.70	<0.05
Minas Gerais	71.64	84.76	97.88	5.35	<0.05
Espírito Santo	52.36	65.46	78.56	6.57	<0.05
Rio de Janeiro	35.37	49.51	63.66	10.69	<0.05
São Paulo	27.33	37.22	47.10	9.42	<0.05
Paraná	56.72	67.38	78.05	6.57	<0.05
Santa Catarina	74.66	86.33	98.00	4.47	<0.05
Rio Grande do Sul	37.52	55.60	79.68	11.72	<0.05
Mato Grosso do Sul	60.72	72.38	84.04	5.38	<0.05
Mato Grosso	62.43	73.57	84.70	4.93	<0.05
Goiás	64.67	71.12	77.57	3.03	<0.05
Distrito Federal	7.30	36.52	65.74	35.11	<0.05

## Data Availability

Materials such as protocols, analytical methods, and study materials are available upon request to interested researchers. The authors will make the raw data supporting the conclusions of this manuscript available without undue reservation to any qualified researcher.

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
