# Peer review of "Trends and Geographical Distribution of Family Health Strategy in Brazil from 2009–2023"

_healthcare, 2025, doi:10.3390/healthcare13111246_

Round 1

Reviewer 1 Report

Comments and Suggestions for Authors

The manuscript presents an analysis of the coverage of the Family Health Strategy (FHS) in a country for 15 years with an objective to make a correlation to the Human Development Index (HDI). The authors took  data of total family health teams, total population, and HDI. The researchers used  the Prais-Wisten regression and linear regression to find out FHS in the country. 

The manuscript is well organized and presented. However, it lacks the followings:

The major issue with the paper is the lack of highlighting the main contributions of the work. What was the main importance of the work that was demanded to conduct this research. As it is observed that think kind of study and analysis is performed/required in publication institutions when a long/short planning is done. So, what was motivation of the underlying research and what is distinctive contribution of the proposal?

I was interested to see any worthwhile suggestions proposed by the study. As described in the document that "five federative units did not show a statistically significant increase in FHS coverage", and Linear regression showed a significant negative correlation between HDI values and FHS coverage", etc. So, how these challenges could be targeted and resolved? Likewise, authors pointed out some limitations of the work in the end, how these limitations effect the study?

The authors cited many Webites without giving the date of consultation. So, authors should provide the date of consultation too. Equations used in the manuscript should also be cited properly and explained accordingly in terms of notations and variables.

Author Response

Reviewer#1

The manuscript presents an analysis of the coverage of the Family Health Strategy (FHS) in a country for 15 years with an objective to make a correlation to the Human Development Index (HDI). The authors took  data of total family health teams, total population, and HDI. The researchers used  the Prais-Wisten regression and linear regression to find out FHS in the country. 

The manuscript is well organized and presented. However, it lacks the followings:

The major issue with the paper is the lack of highlighting the main contributions of the work. What was the main importance of the work that was demanded to conduct this research. As it is observed that think kind of study and analysis is performed/required in publication institutions when a long/short planning is done. So, what was motivation of the underlying research and what is distinctive contribution of the proposal?

R: We thank this reviewer for his/her attentive reading of our manuscript and the support of our work. We understand this reviewer’s concerns, and we amended the text as appropriate to reach his/her requests/suggestions. We included additional text in the Introduction (“motivation”) and the Discussion sections (“contribution”).

Sentences added to the first and second paragraphs of the Introduction help in additionally contextualizing the study. Several new sentences were added to practically all paragraphs in the Discussion section.

I was interested to see any worthwhile suggestions proposed by the study. As described in the document that "five federative units did not show a statistically significant increase in FHS coverage", and Linear regression showed a significant negative correlation between HDI values and FHS coverage", etc. So, how these challenges could be targeted and resolved? Likewise, authors pointed out some limitations of the work in the end, how these limitations effect the study?

R: We understand this reviewer’s concerns and tried to clarify the two points raised by him/her. Therefore, additional text was included in the Discussion section. As far as possible, we included additional explanations and thoughts in the Discussion section. For this purpose, we added sentences to the discussion, particularly in the second and third paragraphs of the Discussion section, including new references (18-23).

The authors cited many Webites without giving the date of consultation. So, authors should provide the date of consultation too. Equations used in the manuscript should also be cited properly and explained accordingly in terms of notations and variables.

R: We agree with this comment and made the appropriate corrections and amendments to several references.

Reviewer 2 Report

Comments and Suggestions for Authors

Thank you for the opportunity to review this manuscript. It concerns the empirical verification of the implementation of systemic assumptions of the health service in Brazil related to its accessibility for every citizen.

First of all, I would like to appreciate the goal set by the authors, because the verification of the assumptions of social policy in the field of services/benefits should be systematically implemented.

The authors write in lines 80-82 that the goal of this study is to update the coverage of the FHS in Brazil and to examine whether the distribution of resources solves the problem of inequality. At the same time, they inform that this will be done by verifying the coverage of the FHS in relation to the HDI index. Then they indicate that the negative correlation confirms the principle of equality. So, can the stated goal of the research be finally verified on this basis? In my opinion, the authors should use other, additional methods in such a case. For this purpose, convergence analysis could be used, and I recommend such a solution.

Additionally, I recommend several further improvements related to the text itself. First of all, please expand on a few abbreviations (such as CAGR in line 233) or threads, such as the Alma-Alta declaration in line 54.

It will be clearer for the reader to explain why the negative correlation confirms the purpose of the work (such a laconic statement appears both in the last sentence of the abstract and the text itself.

Moreover, the discussion section and, above all, conclusions should be expanded. The work should include more references (discussions with them) to the works cited in lines 255-260. The last part should contain recommendations, especially since positive FHS coverage was achieved in most territorial units, hence the path to achieving more favorable results for the area (Amapa) may be clearer.

Author Response

Reviewer#2

Thank you for the opportunity to review this manuscript. It concerns the empirical verification of the implementation of systemic assumptions of the health service in Brazil related to its accessibility for every citizen.

First of all, I would like to appreciate the goal set by the authors because the verification of the assumptions of social policy in the field of services/benefits should be systematically implemented.

R: We thank this reviewer for his/her appreciation of the manuscript and the support of our work.

The authors write in lines 80-82 that the goal of this study is to update the coverage of the FHS in Brazil and to examine whether the distribution of resources solves the problem of inequality. At the same time, they inform that this will be done by verifying the coverage of the FHS in relation to the HDI index. Then they indicate that the negative correlation confirms the principle of equality. So, can the stated goal of the research be finally verified on this basis? In my opinion, the authors should use other, additional methods in such a case. For this purpose, convergence analysis could be used, and I recommend such a solution.

R: We appreciate this reviewer’s comment and attentive reading of our manuscript. We expanded our analysis of the subject, adding more text and references to the Discussion section, as suggested. Several insertions were made to all paragraphs in the Discussion section.

Additionally, I recommend several further improvements related to the text itself. First of all, please expand on a few abbreviations (such as CAGR in line 233) or threads, such as the Alma-Alta declaration in line 54.

R: We agree with this comment and made the suggested amendments (line 368 of the marked-up manuscript).

It will be clearer for the reader to explain why the negative correlation confirms the purpose of the work (such a laconic statement appears both in the last sentence of the abstract and the text itself.

R: We understand this reviewer’s concern and amended the text as appropriate, attempting to clarify the idea in the new Discussion section and the new abstract. We added several new sentences to the manuscript with the respective complementary references. The respective insertions appear mainly in the fourth paragraph, including new references 26 and 27.

Moreover, the discussion section and, above all, conclusions should be expanded. The work should include more references (discussions with them) to the works cited in lines 255-260. The last part should contain recommendations, especially since positive FHS coverage was achieved in most territorial units, hence the path to achieving more favorable results for the area (Amapa) may be clearer.

R: We appreciate this reviewer’s support and have followed his/her suggestions, including additional text and the respective references. Several new references were included in the Discussion section, and the Conclusions were completely rewritten.

Additional references to the Discussion section:

  • Macinko et al., J Epidemiol Community Health. 2006;60(1):13-19;
  • Giovanella et al, Ciênc Saúde Coletiva. 2021; 26:2543–56;
  • Arantes et al., Cien Saude Colet. 2016;21(5):1499-1510;
  • Russo et al., Cien Saude Colet. 2021;26(4):1585-1594;
  • Girardi et al., Cien Saude Colet. 2016;21(9):2675-2684;
  • Hone et al., BMC Health Serv Res. 2020;20(1):873;
  • Souza Júnior et al., Cien Saude Colet. 2021;26(suppl 1):2529-2541;
  • Sarti et al., Epidemiol Serv Saude. 2020;29(2):e2020166;
  • Rawaf et al., Eur J Gen Pract. 2020;26(1):129-133;
  • Andrade et al., PLoS One. 2021;16(5): e0251764;

Reviewer 3 Report

Comments and Suggestions for Authors

The manuscript entitled “Trends and geographical distribution of Family Health Strategy in Brazil from 2009–2023” presents an interesting and comprehensive analysis of the Family Health Strategy (FHS) coverage trends in Brazil.

Strengths:

  • The topic is timely and relevant, particularly for health policy researchers.
  • The use of Prais-Winsten regression and the evaluation of correlation with HDI are appropriate and statistically sound.
  • The article is clearly written and logically structured.

Suggestions for improvement:

  • In the Methods section, more detailed information could be provided regarding how missing HDI data were imputed for different states.
  • The authors could specify if and how the influence of the COVID-19 pandemic (2020–2022) was considered, as it could have impacted FHS coverage dynamics.
  • The Discussion could be further strengthened by explicitly comparing findings with previous similar studies beyond those cited.

Overall, the study is well-conducted and contributes important insights into healthcare equity in Brazil. Only minor improvements are recommended.

Author Response

Reviewer#3

The manuscript entitled “Trends and geographical distribution of Family Health Strategy in Brazil from 2009–2023” presents an interesting and comprehensive analysis of the Family Health Strategy (FHS) coverage trends in Brazil.

Strengths:

  • The topic is timely and relevant, particularly for health policy researchers.
  • The use of Prais-Winsten regression and the evaluation of correlation with HDI are appropriate and statistically sound.
  • The article is clearly written and logically structured.

Suggestions for improvement:

  • In the Methods section, more detailed information could be provided regarding how missing HDI data were imputed for different states.

R: We thank this reviewer for his/her appreciation of our study and support of the manuscript. We understand his/her concerns regarding the missing data and included additional information as suggested, with a new reference (Reference#15: Enders, C.K. Applied Missing Data Analysis. The Guilford Press, New York., USA, 2010).

  • The authors could specify if and how the influence of the COVID-19 pandemic (2020–2022) was considered, as it could have impacted FHS coverage dynamics.

R: We understand this reviewer’s concern regarding the pandemic period. Although the global emergency had broad impacts on scientific research, particularly changing priorities in the medical field, practical issues and data collection challenges related to the pandemic had a relatively minor effect on this specific work. Analyzing the data collected for this study, we did not detect any abrupt or unexpected change in the numbers obtained before and after the pandemic period. This can be observed, for example, when analyzing Figure 1, which shows a linear increase in numbers, and Figure 3, which shows a linear increase in the years 2021 and 2023. Nonetheless, we added sentences to the fifth paragraph of the Discussion.

The Discussion could be further strengthened by explicitly comparing findings with previous similar studies beyond those cited.

R: We agree with this comment and included additional sentences and comparisons with our study. We included several new references (new references #18-23; #26; and #28-30) with comments and comparisons with our results in the Discussion section, as far as possible.

Overall, the study is well-conducted and contributes important insights into healthcare equity in Brazil. Only minor improvements are recommended.

R: Again, we thank this reviewer for his/her suggestions and support for our study.

Round 2

Reviewer 1 Report

Comments and Suggestions for Authors

The manuscript presents an analysis of the coverage of FHS with an objective to make a correlation to HDI. The authors took  data of total family health teams, total population, and HDI. The researchers used  the Prais-Wisten regression and linear regression.

I already reviewed the manuscript and give feedback. I really appreciate the authors who consider my comments and update the document. 

I commented to point out the motivation, need, and main contributions that are provided by the researcher in the document. The contributor also explained suggestions and recommendations against the highlighted limitations. The consultation dates of cited websites are also provided in the paper. 

I have the perception that the manuscript meets the merit of acceptance. 

Author Response

The manuscript presents an analysis of the coverage of FHS with an objective to make a correlation to HDI. The authors took  data of total family health teams, total population, and HDI. The researchers used  the Prais-Wisten regression and linear regression.

I already reviewed the manuscript and give feedback. I really appreciate the authors who consider my comments and update the document. 

I commented to point out the motivation, need, and main contributions that are provided by the researcher in the document. The contributor also explained suggestions and recommendations against the highlighted limitations. The consultation dates of cited websites are also provided in the paper. 

I have the perception that the manuscript meets the merit of acceptance. 

R: We thank this reviewer for his/her attentive reading of our manuscript and the support of our work. We understand that the reviewer’s concerns have been addressed with all the amendments made in response to his/her requests and suggestions. To clarify the points raised by the reviewers, we added several new sentences to practically all paragraphs in the Discussion section, with respective new references.

We again thank all three reviewers for their attentive reading of our manuscript and for allowing us to improve our work. We believe that the changes made continue to support our study, and the new version is even more consistent and clear following the suggested modifications.